# ResPerfNet: Deep Residual Learning for Regressional Performance Modeling of Deep Neural Networks

## Abstract

The rapid advancements of computing technology facilitate the development of diverse deep learning applications. Unfortunately, the efficiency of parallel computing infrastructures varies widely with neural network models, which hinders the exploration of the design space to find high-performance neural network architectures on specific computing platforms for a given application. To address such a challenge, we propose a deep learning-based method, *ResPerfNet*, which trains a residual neural network with representative datasets obtained on the target platform to predict the performance for a deep neural network. Our experimental results show that ResPerfNet can accurately predict the execution time of individual neural network layers and full network models on a variety of platforms. In particular, ResPerfNet achieves 8.4% of mean absolute percentage error for LeNet, AlexNet and VGG16 on the NVIDIA GTX 1080Ti, which is substantially lower than the previously published works.

## 1 Introduction

Deep learning (DL) has exploded successfully and is applied to many application domains, such as image recognition and object detection Thus, a lot of human experts design high-accuracy neural network architectures for different applications. However, for Internet of Things (IoT) applications, large neural network models cannot fit into resource-constrained devices. On the other hand, a system designer often tries to find a proper computing platform or a deep learning accelerator (DLA) to execute a DL application with acceptable responsiveness. An exhaustive way to optimize the system design is to evaluate the cost and performance of desired DL models on all the available hardware/software options, but it is not only tedious but costly and lengthy in practice.

Since DL frameworks and accelerators are evolving rapidly, and even some slight changes could significantly impact the performance of DL applications, it may be necessary to update the performance models frequently. Therefore, we need a systematic and efficient approach to produce accurate performance models when changes occur. While several works (Qi et al.; Justus et al. (2018); Wang et al.) have been proposed to estimate the delivered performance of a given DL model on a specific computing platform, so as to rapidly evaluate design alternatives, the estimates from these efforts are not very accurate. For example, the mean absolute percentage error (MAPE) for estimating full neural network models such as LeNet (LeCun et al. (1998)), AlexNet (Krizhevsky et al. (2012)) and VGG16 (Simonyan & Zisserman) on the NVIDIA GTX 1080Ti is as high as 24% in Wang et al., whose accuracy is the best among the previous works, but still has room for improvement.

In this paper, we propose a deep residual network architecture, called ResPerfNet, to efficiently and accurately model the performance of DL models running on a wide range of DL frameworks and DLAs. It is based on the residual function approach proposed by (He et al. (2016) and inspired by the prior works Liu & Yang (2018); Jha et al. (2019); Wan et al. (2019)), which use residual neural networks to solve regression problems. The proposed model can be trained with performance data collected from many system configurations to establish a unified performance predictor which assists the users in selecting the DL model, the DL framework, and the DLA for their applications. Extensive experiments have been done to show that our unified approach not only provides more accurate performance estimates than the previous works, but also enables the users to quickly pre-

dict the performance of their DL applications executed with various models-framework-accelerator configurations. The contributions of this paper are summarized as follows.

- An unified DL-based approach for estimating the computing performance of DL applications on a variety of models-framework-accelerator configurations, which enables the users to explore the hardware/software design space quickly.

- A novel deep residual neural architecture is proposed to deliver the most accurate performance predictions that we are aware of. Experimental results confirm that our approach yields lower prediction errors on across various platforms.

The remaining of this paper is organized as follows. Section 2 presents the related work. Section 3 describes the architecture of ResPerfNet. Section 4 shows the proposed systematic modeling method. Section 5 elaborates the dataset and training mechanism to train the ResPerfNet models within a reasonable time span. Section 6 evaluates the efficiency of our approach. Section 7 concludes the paper.

## 2 BACKGROUND AND RELATED WORK

With the rapid evolving of both hardware accelerators and DL models, the performance measure/estimation of the DL models on the DLA platforms is an important task to evaluate the effectiveness of the software/hardware solutions to the given problems. Different approaches have been proposed to serve the purposes.

**Benchmarking approaches**, such as DAWNbench (Coleman et al. (2017)) and MLPerf (Reddi et al. (2020)), aim at the measurements of the training and inference performance of the machine-learning (ML) models on certain software/hardware combinations. By offering a set of standardized machine learning workloads and the instructions for performance benchmarking, these benchmarks are able to measure how fast a system can perform the training and inference for ML models.

**Analytical approach**, as reported in PALEO (Qi et al.), constructs the analytical performance model for DL systems. The execution time is decomposed into the total time for the computation and communication parts, which are derived from the utilization of the computing and communication resources on the target hardware, respectively. For instance, the computation time is estimated by dividing the total floating-point operations required by the DL model to the actual processing speed (i.e., the processed floating-point operations per second for the DL model) delivered by the computing hardware. The communication time is calculated by the similar approach.This approach highly relies on the accuracy of the benchmarking results (i.e., to provide the actual processing speed of the target model on the hardware), which requires its users to choose the benchmarks wisely to perfectly match the program characteristics of their target deep learning models, so as to give a proper estimate of the actual processing speed. However, the manual process (of the benchmarks selection) limit its widespread adoption.

**DL-based approaches** build the DNNs for estimating the DL models' performance by learning the relationships between the characteristics of the DL models and the specifications of the accelerating hardware. The following works focus on TensorFlow-based DL models. Justus et al. (2018) use a fully-connected multiple-layer perceptron (MLP) network for performance prediction, using the configurations of the DL model and the specification of the hardware accelerator, and the training data of the DL model as the input features to the MLP network. However, due to the simplified communication time estimation model, where the communications from GPU to CPU for each of the DL layers are counted repeatedly for estimating the communication time, their model tends to provide over-estimated results. Wang et al. use PerfNet (an MLP network) to learn the relationships between the configurations and the execution time of the target DL model. They further decompose the execution of a DL model into three phases, preprocessing, execution, and postprocessing, and train multiple PerfNet network instances, each of which learns the relationships between the model configurations and the model execution time for a specific phase. By aggregating the prediction results for the three phases, their proposed work is able to predict the total execution time of a given DL model. Nevertheless, the MLP network has its own limitation, i.e., it is hard to further enhance its performance since a deeper MLP network will lead to lower prediction accuracy.

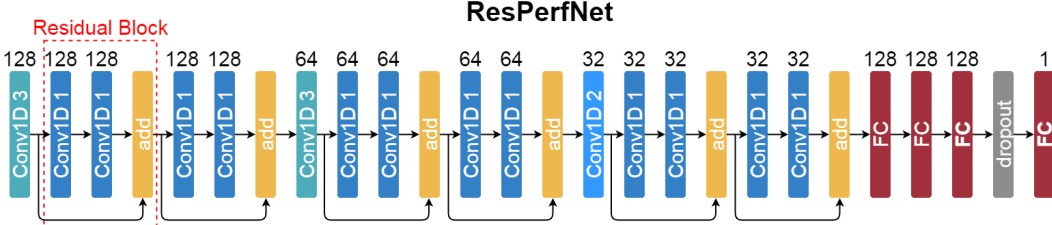

Figure 1: Network architecture of ResPerfNet for performance estimation, where the number above each layer is the kernel number or the output neuron number for the corresponding layer.

In consideration of the limitations of the prior works listed above and the need of modeling the optimizing DL frameworks, our work uses the systematical approach to characterize the DL models built with various DL framework, and adopts the residual neural network to model their delivered performance on the DLAs.

## 3 RESPERFNET ARCHITECTURE

ResPerfNet adopts a ML-based approach for the performance estimation of different types of neural network layers. Furthermore, ResPerfNet is specially designed to prevent the *degradation* problem, which refers to the phenomenon that increasing the depth and/or the width of each layer for the DNN may not only necessarily improve the accuracy, but get saturated rapidly and then degrades sharply as reported in (He & Sun (2015); Srivastava et al. (2015)). In other words, it is more likely to lead to a higher training error on the neural network with a wider or deeper architecture. To solve the problem, the deep residual learning is proposed and applied to each group of the stacked NN layers (He et al. (2016)), where a certain number of stacked layers are logically grouped together to form a *residual block*. Hence, in this work, to address the degradation problem, we adopt the deep residual learning to every few stacked layers (He et al. (2016)).

The residual block is defined as Equation 1, where $\mathbf{x}$ and $\mathbf{y}$ represent the input feature maps and the output vectors of the residual layer, respectively. The function $\mathcal{F}(\mathbf{x}, \{W_i\})$ performs the residual operations to be learned. The operation $\mathcal{F}(\mathbf{x}, \{W_i\}) + \mathbf{x}$ is performed by a shortcut connection and element-wise addition. Figure 1 illustrates the network architecture of ResPerfNet. The second, third and fourth layers (i.e., two convolutional and one add layers) together form a residual block, and there are a total of six residual blocks in ResPerfNet.

$$\mathbf{y} = \mathcal{F}(\mathbf{x}, \{W_i\}) + \mathbf{x} \tag{1}$$

As shown in Figure 1, the ResPerfNet consists of 26 layers, including 15 convolutional layers, 6 add layers, 4 fully-connected (FC) layers and 1 dropout layer. Before FC layers, every 7 layers contain one head convolutional layer (e.g., `Conv1D 3` representing the head convolutional layer for the first residual block) and two residual blocks, each of which consists of two convolutional layers with the same filters and an element-wise add residual function layer. The first head convolutional layer has 128 filters of kernel size 3 with a stride length of 1. In order to reduce the complexity of ResPerfNet, the second head convolutional layer uses 64 filters of kernel size 3 with a stride length of 1. Moreover, the number of filters for the six residual blocks is decreasing from 128 filters in the first two blocks to 32 filters for the last two blocks. Three FC layers are attached to the last residual block, where each of the FC layers has 128 neurons. The dropout layer with the ratio of 0.2 is connected to the last FC layer, which uses a single neuron to perform the one-dimensional regression for predicting the elapsed time of the designated type of the layers.

Our proposed residual neural architecture, ResPerfNet, gets significant improvements in accuracy compared with traditional machine learning algorithms, such as support vector regression, polynomial regression and XGBoost, and is even better than the MLP network. A series of experiments has been done to show ResPerfNet is superior to the previous works in Section 6.1.

## 4 METHODOLOGY

This section presents the methodology of using ResPerfNet to relate the performance characteristics of a CNN layer to the delivered performance of the given layer. We first define the target neural networks for the performance modeling in Section 4.1. The three-phase based modeling of a given CNN based is presented in Section 4.2. Lastly, the same modeling for a given NN layer is further described in Section 4.3.

### 4.1 FORMALIZING THE NEURAL NETWORKS

A neural network can be represented by a directed acyclic graph, denoted as $\mathcal{N}(\{u^{(i)}\}_{i=1}^{k})$, consisting of an ordered sequence of $k$ nodes, where each graph node $u^{(i)}$ represents a layer of the neural network $\mathcal{N}$, such as convolutional, pooling, and fully-connected layers. The input and output feature maps of a graph node $u^{(i)}$ performing the operation $f^{(i)}$ are denoted as $input(f^{(i)})$ and $output(f^{(i)})$, respectively. In this work, we assume that a given neural network will be run on the host system $h$ with a single hardware accelerating device $d$.

### 4.2 THE THREE-PHASE PERFORMANCE MODELING

The execution time of a given neural network model includes the computation time spent on the acceleration device $d$ and the data communication time between the host system $h$ and the device $d$. As most of the computations are performed by the accelerating device and the communications occur merely at the first and the last layers of the given model, the estimated execution time of a given neural network model with $k$ layers is formulated as follows, where the formulation assumes that all $k$ layers within the given model are accelerated by the single device $d$.

$$T(\mathcal{N}) = T_{pre}(u^{(1)}) + \sum_{i=1}^{k} T_{exe}(u^{(i)}) + T_{post}(u^{(k)}) \tag{2}$$

The above equation shows the *three-phase* performance modeling approach, where $T_{pre}$, $T_{exe}$, and $T_{post}$ represent the execution time for the *preprocess*, *execution*, and *postprocess* phases, respectively. Specifically, the communication time of bringing the input data from the host system to the accelerating device at the first layer is denoted as $T_{pre}(u^{(1)})$, where the $i$-th NN layer is represented as $u^{(i)}$. The summation of the execution time for all the NN layers is represented as $\sum_{i=1}^{k} T_{exe}(u^{(i)})$. The communication time of transferring the inference results from the accelerating device to the host system is defined as $T_{post}(u^{(k)})$. Our prediction model delivers more accurate performance estimates than previously proposed methods by modeling these three phases defined in the following subsection for a DLA separately and adding the predicted results together as Equation 2.

### 4.3 MODELING INDIVIDUAL NN LAYERS

The similar approach is used to model the performance of the $i$-th NN layer $u^{(i)}$. In particular, for each layer $u^{(i)}$, the execution times for the preprocess, execution, and postprocess phases are $T_{pre}(u^{(i)})$, $T_{exe}(u^{(i)})$, and $T_{post}(u^{(i)})$, respectively. The above time components constitute the estimated execution time of the layer $u^{(i)}$, as defined in the equation below. The superscript index $i$ is omitted to simplify the looks of the equations by using the simpler form $u$.

$$T(u) = T_{pre}(u) + T_{exe}(u) + T_{post}(u) \tag{3}$$

The preprocess phase is for preparing the input data for the acceleration in $d$ and involves with the four operations: 1) issuing the commands for copying input feature maps on $h$ and $d$ asynchronously, 2) performing the memory copy of the input feature maps in 1, 3) issuing the commands for the operation $f$ on $d$, and 4) performing the data reshaping operations for input feature maps. The data reshaping operations, which transform the input/output data to the more efficient format for the next operation on $d$, usually occur in data transmissions between $h$ and $d$. The lengths of time for the four operations are $\mathcal{R}(input(f), h, d)$, $\mathcal{M}(input(f))$, $\mathcal{R}(f, d)$, and $\mathcal{T}(input(f), d)$, respectively. As shown in Equation 4, the time consumed in the preprocess phase is defined as the summation of the time required by the above four operations.

$$T_{pre}(u) = \mathcal{R}(input(f), h, d) + \mathcal{M}(input(f), h, d) + \mathcal{R}(f, d) + \mathcal{T}(input(f), d) \qquad (4)$$

Intuitively, the time consumed for computation, which is $\mathcal{C}(f, d)$, in the execution phase would be identical to the computation time of $f$ on $d$. Unfortunately, the measured execution time of a layer from the micro-benchmarks includes the time consumed by the data reshaping operations in both directions, from $h$ to $d$ and from $d$ to $h$, which are $\mathcal{T}(input(f), d)$ and $\mathcal{T}(output(f), d)$, respectively. As the deployed NN layers collectively run on the acceleration device $d$, isolating the data reshaping time from the measured execution time for the NN layer of each micro-benchmark facilitates the execution time estimation of the deployed NN layers with the formula, $\sum_{i=1}^{k} T_{exe}(u^{(i)})$. Regarding this situation, the time for the execution phase is defined in Equation 5.

$$T_{exe}(u) = \mathcal{C}(f, d) - \mathcal{T}(input(f), d) - \mathcal{T}(output(f), d) \qquad (5)$$

The postprocess phase is defined for dealing with the procedure of returning the inference computation results back to the invoking application on the host system. That is, it is about reshaping the output vector into the format accepted by $h$, copying the output vector back to $h$ from $d$, and moving the prediction result to the application level (i.e., the call site of the model inference) on the host system. The corresponding execution time for the above three operations are denoted as $\mathcal{T}(output(f), d)$, $\mathcal{M}(output(f, d, h)$, and $\mathcal{V}(output(f), h)$, respectively.

$$T_{post}(u) = \mathcal{T}(output(f), d) + \mathcal{M}(output(f), d, h) + \mathcal{V}(output(f), h) \qquad (6)$$

## 5 TRAINING DATA AND LOSS FUNCTION

In this section, we present the details of the dataset used to build the proposed performance prediction models. In particular, the configurations of our developed benchmark tools for the training dataset is discussed in Section 5.1. The tool collecting and extracting the data is described in Section 5.2, and the data transformation techniques to facilitate the training convergence is introduced in Section 5.3. The specially designed loss function to better deal with the unbalanced training data is introduced in Section 5.4.

### 5.1 DATA PREPARATION

The training data is the characteristics of the TensorFlow and TensorRT programs and the performance information of the programs running on the target computing hardware, where the proposed model helps correlate the characteristics and their runtimes during the training process. In order to better catch the characteristics of different TensorFlow and TensorRT configurations (i.e., the code patterns, which are considered as the features during the model training process), we have developed a benchmark tool to generate a set of micro-benchmarks, which are actually TensorFlow and TensorRT programs with different configurations for the three types of layers, including convolution, pooling and dense layers. The generation of the micro-benchmarks are done by randomly selecting the configurations for each type of the layer, so as to collect the performance for different configurations. The possible configurations (or features) for all three layer types and their ranges are listed in Table 3. These configurations are actually the function parameters for the three types of layers, which are extracted from TensorFlow 1.13 APIs, including `tensorflow.layers.conv2d`, `tensorflow.layers.maxpooling2d`, and `tensorflow.layers.dense`, and their possible combinations are $7.33 \times 10^{14}$, $7.33 \times 10^{10}$, and $2.14 \times 10^{9}$, respectively. While each micro-benchmark takes at least seconds for the stable and accurate measurements, it is impossible to cover the entire design space with brute force, which requires over $10^{14}$ micro-benchmark runs.

### 5.2 DATA COLLECTION AND DATA EXTRACTION

The data preparation is used to generate the TensorFlow- and TensorRT-based micro-benchmarks. The data collection takes about two weeks for running 100,000 different samples of the TensorFlow micro-benchmarks on the DLAs to collect the performance data. On the other hand, for the TensorRT micro-benchmarks, more than two weeks were spent to optimize and profile the 25,000 different configurations of the TensorRT programs. It is interesting to note that the TensorRT experiments generate large optimized intermediate files, especially for the dense layer, where it requires

more than 5TB of storage space to keep its parameters. Due to the disk space limitation, we select 16,000 out of 25,000 samples to run and profile their performance. For data extraction, our data processing tool filters out the outliers (data with extreme values) before feeding the profiled data for the model training. The total elapsed time of each layer is decomposed into the preprocessing time ($T_{pre}$), the execution time ($T_{exe}$), and the postprocessing time ($T_{post}$), as mentioned in the previous section. In order to test the accuracy of our trained model, the collected samples are split into 80% of the samples as training datasets and 20% as testing datasets.

### 5.3 DATA TRANSFORMATION

Now, suppose we are given a training dataset $\mathcal{D}$, which is comprising $m$ observations and $p$ features of $X$ and written as $\mathcal{D} = \{t_i, x_{i1}, x_{i2}, ..., x_{ip}\}_{i=1}^m$, where $\mathbf{t}$ is a vector of observed values $t_i (i = 1, ..., m)$, and $X$ could be seen as a matrix of row-vectors $\mathbf{x}_i (i = 1, ..., m)$ or of m-dimensional column-vectors $X_j (j = 1, ..., p)$. The coefficients vector $\mathbf{w}$ keeps the weights of the model. The predicted value is denoted as $y(\mathbf{x}, \mathbf{w})$, for any given model of weights $\mathbf{w}$ and the dataset $\mathbf{x}$. In order to improve the convergence efficiency and stability of the stochastic gradient descent (SGD) algorithm, the three types of data transformations are adopted in this work, including scalar multiplication, Z-scores transformation, and Box-Cox transformation. Scalar multiplication is used to provide fine-grained updates of the SGD procedure and scales each observed value $t_i$. Z-scores transformation puts each data feature $X_j$ from different sources into the same scale to eliminate the prejudicial bias of the features values. Box-Cox transformation converts the values of the features $X_j$ to standard normal random variables, which would further improve the effectiveness of Z-scores transformation. Details of these data transformations are available in Appendixes B, C and D.

### 5.4 LOSS FUNCTION

As the observed vector $\mathbf{t}$ is with the positive-skew distribution and often contains some noises contributed by the measurement errors, we fine-tune the loss function as mean absolute percentage logarithmic error (MAPLE) for the prediction model (Wang et al.), as shown in Equation 7. To deal with the situation of the skewed distribution, the logarithmic operations for the predicted values $1 + y(\mathbf{x}_i, \mathbf{w})$ and the observed values $1 + t_i$, and the division operation on the observed values in MAPLE are expected to enhance the accuracy of the small data, which occurs frequently. On the other head, the absolute value of MAPLE helps increase the resistance against outliers that may unexpectedly appear in the measured data. Moreover, to prevent over-fitting, L2 regularization is added to the loss function, where $\lambda_2$ is a scaling factor for the regularization.

$$E_n(\mathbf{w}) = \frac{1}{n} \sum_{i=0}^n \left| \frac{\log(1 + y(\mathbf{x}_i, \mathbf{w})) - \log(1 + t_i)}{\log(1 + t_i)} \right| + \lambda_2 \|\mathbf{w}\|^2 \tag{7}$$

## 6 EVALUATION

The layer-wise and model-wise performance results are evaluated to demonstrate the effectiveness of ResPerfNet in this section. In particular, we compare the layer-wise estimated execution time produced by ResPerfNet and the previous works to show that ResPerfNet is superior to other regression based approaches, such as polynomial regression, support vector regression and PerfNet. Three statistical metrics, including mean absolute percentage error (MAPE), root mean squared error (RMSE) and mean absolute error (MAE), are used to quantify the effectiveness for each tested performance modeling approach. In addition, to demonstrate the capability of ResPerfNet for the full model prediction, three popular CNNs are considered in the model-wise experiments, e.g., LeNet, AlexNet, and VGG16. Note that three data transformations mentioned in Section 5.3 are applied in ResPerfNet by default unless specified otherwise. The details of our experimental environments are listed in Appendix F.

### 6.1 LAYER-WISE EXECUTION TIME PREDICTION

Table 1 compares the MAPEs of the execution time for the convolutional layers, estimated by ResPerfNet and the prior works. While appropriate parameter adjustments are applied to obtain parame-

ters for better results, the MAPEs of polynomial regression, support vector regression, and XGBoost are over 29%, which means the error is quite large and indicates that the corresponding approaches are not capable of doing good performance prediction for the real applications. On the contrary, the DL-based approaches, PerfNet and ResPerfNet, give more accurate estimations, which have less than 15% of the MAPEs. In particular, ResPerfNet outperforms the other approaches and has 11.75% and 14.23% of the MAPEs for the TensorFlow and TensorRT models. The results suggest that ResPerfNet correctly associates the program characteristics to the performance model.

Table 1: Comparison of the prediction errors (MAPEs) for the convolutional layer execution time produced by different approaches.

| Framwork | Layer | Phases | Polynomial Regression | SVR RBF | XGBoost | PerfNet | ResPerfNet BoxCox |
|---|---|---|---|---|---|---|---|
| TensorFlow | convolutional | execution | 63.57 % | 58.74 % | 29.25 % | 14.96 % | 11.75 % |
| TensorRT | convolutional | execution | 316.5 % | 59.94 % | 33.10 % | - | 14.23 % |

To further look into the effectiveness of PerfNet and ResPerfNet and the impact of the Box-Cox transformation on the predicted results, Figure 2 plots the error curves of the TensorFlow convolutional layer using the PerfNet and ResPerfNet with and without performing the data transformation. Figure 2(a) shows that most of the MAPE of ResPerfNet on the testing dataset are below 15%, as depicted by the red/black solid lines. Notably, ResPerfNet applying the Box-Cox data transformation reaches the lowest prediction error (11.7%), 2% less than ResPerfNet without the data transforming. Similar trends can be observed in Figure 2(b) using the RMSE metric, in which the black solid line also shows the best performance. The results presented in Figure 2 show that ResPerfNet with Box-Cox transformation has better convergence rate, given the same training epoch. The detailed training process is illustrated in Appendix E. Moreover, the $R^2$ values for the predicted and measured execution time of the convolutional, pooling and dense layers are all above 0.97, which demonstrate high prediction quality of ResPerfNet, as illustrated in Figure 4 of Appendix I.

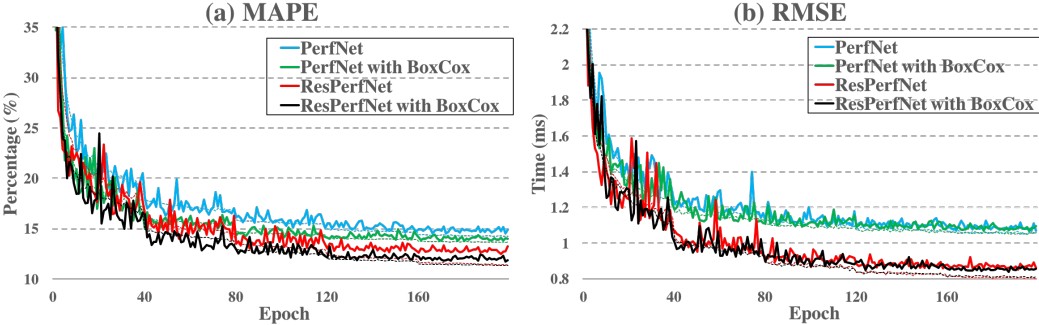

Figure 2: Training and testing errors of PerfNet and ResPerfNet on their TensorFlow models, where the dashed lines represent the training errors, and the solid lines denote the testing errors, with respect to (a) MAPE and (b) RMSE.

The layer-wise performance results of the TensorFlow and TensorRT models delivered by ResPerfNet are listed in Table 2. Overall, the MAPE for all phases are under 16%, which removes the concern of over-fitting. For RMSE, the value of the TensorFlow version convolutional layer is 0.84ms. It is better than the 0.98ms reported by PerfNet (Wang et al.), and is also better than the 2.55ms produced by the method in (Justus et al. (2018)). Detailed predicted results of the three layers under different phases for TensorFlow (with 3 additional platforms) and for TensorRT are also presented in Appendixes G and H, respectively. From the tables, we can see that ResPerfNet has better predicted results for TensorFlow than TensorRT. That is because currently the TensorRT-based ResPerfNet is trained with less training data, as described in Section 5.2. We believe that the accuracy for TensorRT predictions can be further improved with sufficient data as TensorFlow.

Table 2: ResperfNet: TensorFlow/TensorRT predicted inference results on NVIDIA GTX 1080Ti.

| Framwork | Layers | Phases | MAPE (%) | RMSE (ms) | MAE (ms) |
|---|---|---|---|---|---|
| TensorFlow | convolutional | execution | 11.75 | 0.840 | 0.336 |
| | pooling | execution | 6.221 | 0.367 | 0.125 |
| | dense | execution | 4.515 | 0.069 | 0.036 |
| TensorRT | convolutional | execution | 14.23 | 0.649 | 0.263 |
| | pooling | execution | 15.19 | 0.497 | 0.186 |
| | dense | execution | 13.40 | 0.094 | 0.045 |

## 6.2 MODEL-WISE EXECUTION TIME PREDICTION

Figure 3 plots the inference time estimated by PerfNet and ResPerfNet for the three popular DNNs, including LeNet, AlexNet, and VGG16, using TensorFlow and TensorRT frameworks. Figure 3(a)-(c) shows that ResPerfNet has more accurate estimation than PerfNet since the averaged MAPE of the three models is 8.4% for all tested batch sizes, while PerfNet has the averaged MAPE of 24.04%. Figure 3(d)-(f) illustrates the similar trend for TensorRT based DNNs. The averaged MAPE of these DNNs using ResPerfNet is 17%. The results show that our modeling and methodology are effective on the two popular frameworks.

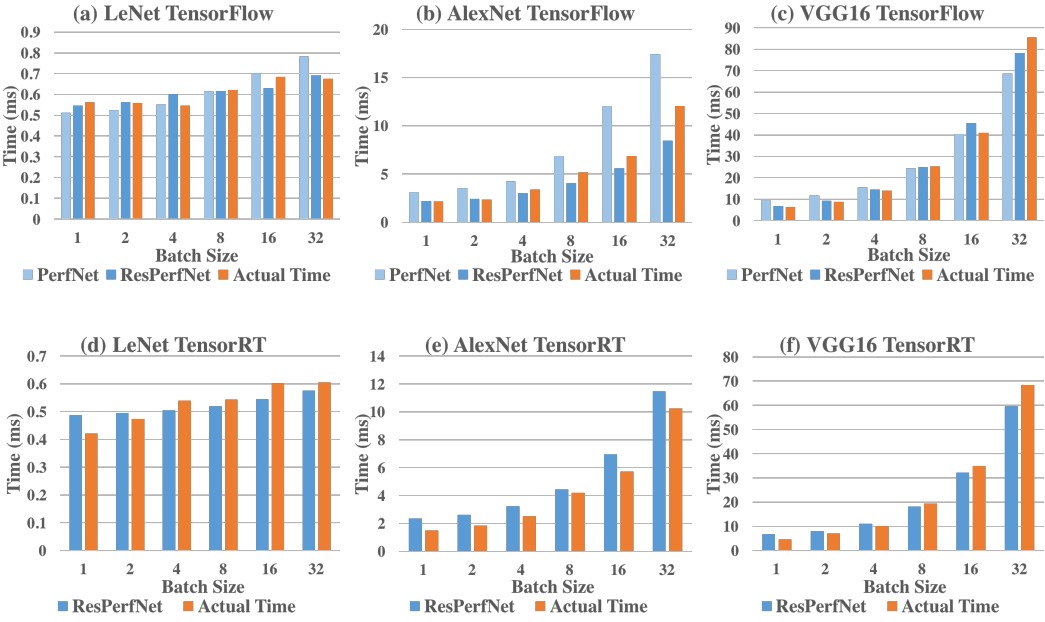

Figure 3: Inference Time Prediction on NVIDIA GTX 1080Ti: Comparison with actual inference time for (a) LeNet on Tensorflow, (b) AlexNet LeNet on Tensorflow, (c) VGG16 on Tensorflow, (d) LeNet on TensorRT, (e) AlexNet on TensorRT, and (f) VGG16 on TensorRT.

## 7 CONCLUSION

In this paper, we proposed a deep residual network architecture, ResPerfNet, to model the performance of neural networks on the target DLAs by considering the interactions between the host and the GPU and decomposing a neural network operation into three phases. In addition, we apply ResPerfNet to predict the execution time of the optimized models, such as TensorRT, with the same performance characteristics as those used in unoptimized models. Our experimental results show that ResPerfNet is able to provide high-accuracy estimations on various DLAs, which helps facilitate the exploration of proper neural network architectures built with various DL frameworks.

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

## A  FEATURES OF TRAINING DATA

Table 3: Description of features.

| Name | Description | Range | Scenario[1] |
|---|---|---|---|
| Batch Size | The number of parallel processed in one iteration. | 1-64 | C, D, P |
| Matrix Size | The dimensions of the input data. | (1-512) × (1-512) | C, P |
| Kernel Size | The size of the filter applied to the input data. | (1-7) × (1-7) | C |
| Channel In | The number of channels in the input data. | 1-9999 | C, P |
| Channel Out | The number of channels in the output data. | 1-9999 | C |
| Strides | The amount of the window shifts for each dimension of the input data with kernels. | (1-4) × (1-4) | C, P |
| Padding | The number for preserving the original size of the image while the filter scans each pixel. 0: Valid. 1: Same. | 0-1 | C, P |
| Activate Function | The number for representing what activation function is used. 0: Without an activate function. 1: ReLU. | 0-1 | C, D, P |
| Bias | The boolean number for utilizing an additional intercept on training input data. | 0-1 | C, D |
| Dimension Input | The number of outputs from the previous layer. | 1-4096 | D |
| Dimension Output | The number of outputs of the layer. | 1-4096 | D |
| Pooling Size | The windows size factors for scaling down the input data. | (1-7) × (1-7) | P |

[1] In the scenario column, C, P, and D indicate which of the NN layer (i.e., Convolutional, Pooling, and Dense layer) the corresponding feature applies to.

## B  SCALAR MULTIPLICATION

Scalar multiplication is applied on the observed vector $\mathbf{t}$ as Equation 8 to magnify the prediction results since the original data are too small to provide accurate estimates. It is interesting to note that the scalar multiplication would be inefficient for some commonly used loss function, such as mean squared error MSE, based on our experiences; nevertheless, it works well with the MAPLE by making every gradient converging smoothly without frequently adjusting an appropriate learning rate in each epoch.

$$scalar\_multiplication : \mathbf{t} = \mathbf{t} \times scaler \tag{8}$$

## C  Z-SCORES TRANSFORMATION

Z-scores transformation is performed on each n-dimensional column-vector $X_j$ as Equation 9, where $\bar{X}_j$ is the mean of the each column-vector $X_j$, and $\sigma_j$ is the standard deviation of each column-vector $X_j$. Z-scores transformation resales the values of the features to ensure the mean to be zero and the standard deviation to be one. The values of the features are rescaled within the range between zero and one, which is useful for gradient decent algorithms.

$$Z\text{-}scores\ transformation : X_j = \frac{X_j - \bar{X}_j}{\sigma_j} \tag{9}$$

## D  BOX-COX TRANSFORM

Box-Cox transformation transforms the input features $X_j$ into a normal distribution for the best model accuracy. Box-Cox transformation is shown in Equation 10, where $\lambda_1$ is the best approximation for the selected features. In our experiments, Box-Cox transformation is applied on *Matrix Size* and *Kernel Size* (See Table 3) for the convolutional layer data, and *Matrix Size* for the pooling layer data.

$$Box\text{-}Cox\ transformation:\ X_j^{(\lambda_1)} = \begin{cases} \frac{X_j^{\lambda_1-1}}{\lambda_1} & if\lambda_1 \neq 0 \\ \ln X_j & if\lambda_1 = 0 \end{cases} \tag{10}$$

## E  THE PROPOSED GRADIENT DESCENT ALGORITHM

Algorithm 1 is the pseudo-code of our proposed algorithm to train each phase of the layers. The required parameters are defined as follows: $optimizer$: algorithm used to update the attributes of a neural network, $lr\_scheduler$: sets the learning rate of each parameter group to the initial $lr$ times a given function, $total\_epochs$: total epochs of the neural network algorithm, $lr$: learning rate, $bs$: maximum of the batch size for each epoch, $\eta$: period of learning rate decay, $\gamma$: multiplicative factor of learning rate decay, and $\lambda_2$: multiplicative factor for the weight penalty.

---

**Algorithm 1** The stochastic gradient descent algorithm proposed by ResperfNet, where our default settings for the our DL regression problems are $optimizer = Adam$, $total\_epochs = 200$, $lr = 0.1$, $bs = 128$, $\eta = 40$, $\gamma = 0.5$, $\lambda_2 = 0.1$, and $scaler = 10$.

---
**Require:** $\alpha$: Multiplicative factor for the weight, $n$: Current batch size, $\tau$: Current iteration.
1: $\mathbf{t} \leftarrow scalar\_multiplication(\mathbf{t}, scalar)$        ▷ Update $\mathbf{t}$ by Equation 8.
2: $\mathbf{x} \leftarrow Z\text{-}scores(Box\text{-}Cox(\mathbf{x}))$       ▷ Update $\mathbf{x}$ by Equation 10 and 9.
3: **for** $e$ in $total\_epochs$ **do**
4:    $lr \leftarrow lr\_scheduler(lr, e, \eta, \gamma)$     ▷ Update the learning rate by scheduler.
5:    **for** $b$ in $(m/bs + 1)$ **do**
6:      $\alpha \leftarrow optimizer(lr)$      ▷ Update the weight factor by optimizer.
7:      $n \leftarrow \mathbf{x}[b * bs : \min((b+1) * bs, m)]$    ▷ Calculate $n$ (current batch size).
8:      $\nabla E_n \leftarrow$ calculate gradient of $E_n$ on model $\mathbf{w}^\tau$    ▷ Calculate $E_n$ by Equation 7.
9:      $\mathbf{w}^{\tau+1} \leftarrow \mathbf{w}^\tau - \alpha \nabla E_n(\mathbf{w}^\tau)$
10:     $\tau \leftarrow \tau + 1$
11:    **end for**
12: **end for**

---

## F  EXPERIMENTAL SETUP

The experiments are done on the Intel i7 processors with a variety of hardware accelerators listed in Table 4. TensorFlow 1.13.1 and TensorRT 5.0.2.6 with Python 3.6 are used to build the DL models, running on Ubuntu 18.04.4 LTS (kernel version 5.4.0-42-generic).

Table 4: Acceleration hardware specifications.

| NVIDIA Device | Basic Clock | CUDA Cores | Memory Clock | Memory Bandwith | Peak TFLOPS | Bus Standard |
|---|---|---|---|---|---|---|
| GTX1080Ti | 1481 MHz | 3584 | 1376 MHz | 484.4 GB/s | 11.34 | PCIe |
| P1000 | 1266 MHz | 640 | 1253 MHz | 80.19 GB/s | 1.894 | PCIe |
| P2000 | 1076 MHz | 1024 | 1752 MHz | 140.2 GB/s | 3.031 | PCIe |
| P5000 | 1607 MHz | 2560 | 1127 MHz | 288.5 GB/s | 8.873 | PCIe |

# G LAYER-WISE EXECUTION TIME PREDICTION FOR TENSORFLOW

Table 5: TensorFlow predicted inference results on NVIDIA GTX 1080Ti.

| Layers | Phases | MAPE (%) | RMSE (ms) | MAE (ms) |
|---|---|---|---|---|
| convolutional | preprocess | 1.603 | 0.215 | 0.126 |
| | execution | 11.75 | 0.84 | 0.336 |
| | postprocess | 2.821 | 0.097 | 0.041 |
| pooling | preprocess | 1.177 | 0.150 | 0.094 |
| | execution | 6.221 | 0.367 | 0.125 |
| | postprocess | 1.967 | 0.091 | 0.031 |
| dense | preprocess | 6.455 | 0.036 | 0.025 |
| | execution | 4.515 | 0.069 | 0.036 |
| | postprocess | 13.86 | 0.011 | 0.009 |

Table 6: TensorFlow predicted inference results on NVIDIA Quadro P1000.

| Layers | Phases | MAPE (%) | RMSE (ms) | MAE (ms) |
|---|---|---|---|---|
| convolutional | preprocess | 1.943 | 0.478 | 0.271 |
| | execution | 12.74 | 5.022 | 2.046 |
| | postprocess | 2.818 | 0.105 | 0.038 |
| pooling | preprocess | 1.511 | 0.206 | 0.122 |
| | execution | 5.842 | 0.977 | 0.390 |
| | postprocess | 2.860 | 0.091 | 0.035 |
| dense | preprocess | 6.691 | 0.043 | 0.035 |
| | execution | 5.687 | 0.429 | 0.204 |
| | postprocess | 13.61 | 0.018 | 0.008 |

Table 7: TensorFlow predicted inference results on NVIDIA Quadro P2000.

| Layers | Phases | MAPE (%) | RMSE (ms) | MAE (ms) |
|---|---|---|---|---|
| convolutional | preprocess | 1.535 | 0.536 | 0.308 |
| | execution | 12.347 | 4.209 | 1.671 |
| | postprocess | 3.132 | 0.107 | 0.045 |
| pooling | preprocess | 1.321 | 0.407 | 0.185 |
| | execution | 4.425 | 0.682 | 0.234 |
| | postprocess | 2.728 | 0.098 | 0.034 |
| dense | preprocess | 7.369 | 0.017 | 0.013 |
| | execution | 13.39 | 0.073 | 0.037 |
| | postprocess | 14.03 | 0.011 | 0.008 |

Table 8: TensorFlow predicted inference results on NVIDIA Quadro P5000.

| Layers | Phases | MAPE (%) | RMSE (ms) | MAE (ms) |
|---|---|---|---|---|
| convolutional | preprocess | 1.247 | 0.417 | 0.235 |
| | execution | 12.23 | 1.553 | 0.671 |
| | postprocess | 4.072 | 0.148 | 0.065 |
| pooling | preprocess | 1.339 | 0.398 | 0.205 |
| | execution | 6.444 | 0.527 | 0.174 |
| | postprocess | 3.802 | 0.131 | 0.053 |
| dense | preprocess | 5.126 | 0.045 | 0.035 |
| | execution | 5.298 | 0.121 | 0.121 |
| | postprocess | 18.13 | 0.019 | 0.019 |

# H LAYER-WISE EXECUTION TIME PREDICTION FOR TENSORRT

Table 9: TensorRT predicted inference results on NVIDIA GTX 1080Ti.

| Layers | Phases | MAPE (%) | RMSE (ms) | MAE (ms) |
|---|---|---|---|---|
| convolutional | preprocess | 2.283 | 0.646 | 0.356 |
| | execution | 14.23 | 0.649 | 0.263 |
| | postprocess | 4.334 | 0.274 | 0.089 |
| pooling | preprocess | 2.268 | 0.689 | 0.323 |
| | execution | 15.19 | 0.497 | 0.186 |
| | postprocess | 4.001 | 0.141 | 0.058 |
| dense | preprocess | 5.612 | 0.035 | 0.027 |
| | execution | 13.40 | 0.094 | 0.045 |
| | postprocess | 18.01 | 0.017 | 0.014 |

# I PREDICTED VS. MEASURED TIME (TENSORFLOW)

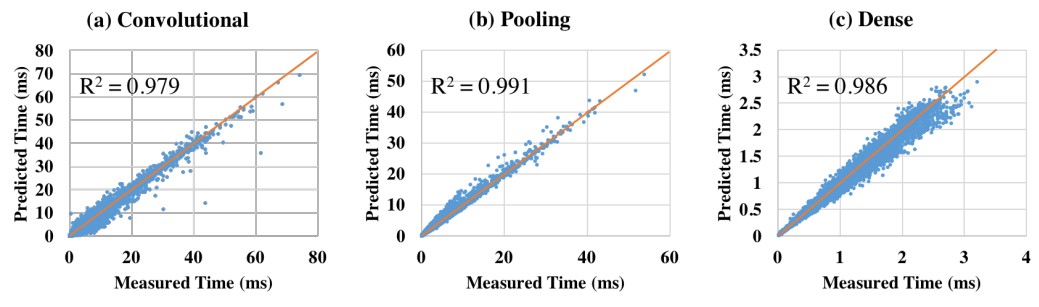

Figure 4: Predicted and measured times of execution phase of (a) Convolutional, (b) Pooling, and (c) Dense layer on a NVIDIA GTX 1080Ti.

# J MODEL-WISE EXECUTION TIME PREDICTION FOR TENSORFLOW

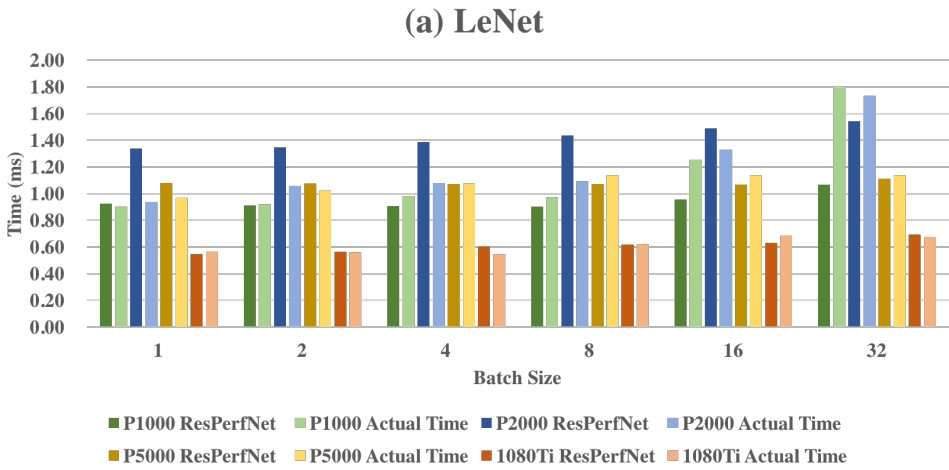

Figure 5: Predicted and actual inference time comparison for LeNet on TensorFlow.

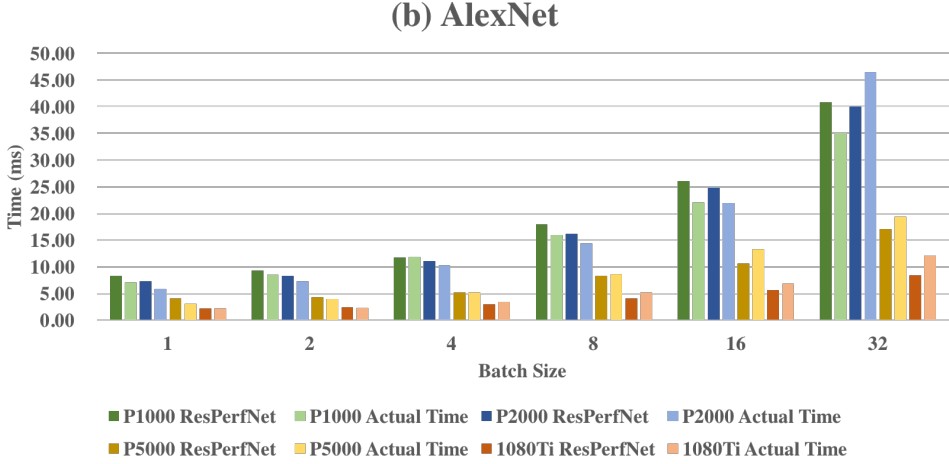

Figure 6: Predicted and actual inference time comparison for AlexNet on TensorFlow.

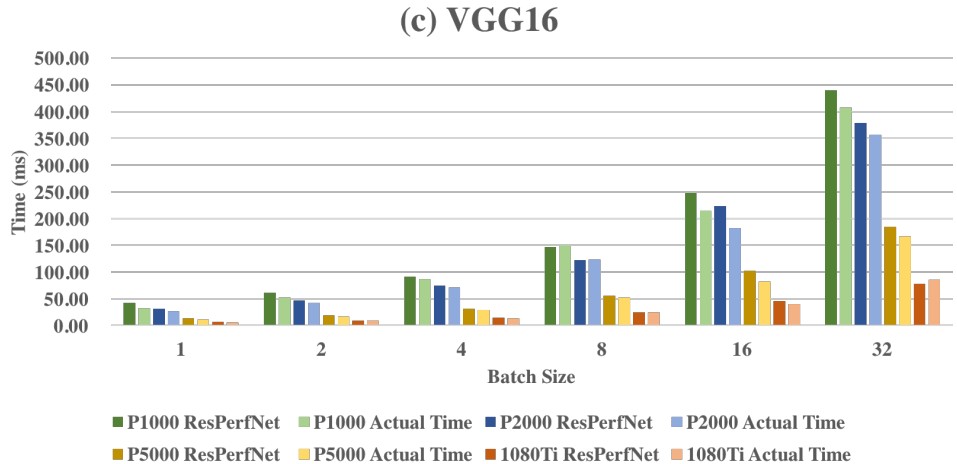

Figure 7: Predicted and actual inference time comparison for VGG16 on TensorFlow.

