# OpenReview forum: "ResPerfNet: Deep Residual Learning for Regressional Performance Modeling of Deep Neural Networks"
_ICLR.cc/2021/Conference — Reject_

### Official Review · AnonReviewer3 · 2020-10-18
**Lack of Noverty and technical contributions**

**Rating:** 4
**Confidence:** 3

**Review:**

Topic

Using a residual-based network to the performance of another DL-based network.

Contributions:

Using a residual-based network for estimating the computing performance of DL applications on a variety of models-framework-accelerator configurations, which enables the users to explore the hardware/software design space.
Using three-phase performance modeling to estimate computation time.

Weakness

1, The main problem is the lack of Novelty and technical contributions. Using a DL-based model to predict the performance is not novel. Many NAS methods using the same trick to estimate the performance of one architecture ahead of running it directly on the hardware.

2, Using a DL-base regression is better than normal regression when the sample size is large. The experiment results are not surprising.

3, Box-cox transformation is a common technique in regression. That is not new.

4, Need to prepare the dataset using a lot of samples, i.e. 100000, which is computationally heavy.
Require huge storage space to keep the samples for even one platform.
Thus the whole method is not efficient. It is hard to generalize it to other hardware/platform.

5, Besides, the method only considers some common operations such as conv, pooling, and FC-layer. However, more operations should be considered for example ROI Pooling, NMS, Spatial-to-depth. Those operations are also commonly used in tasks such as detection and SR.

---

> ### Author Response · Authors · 2020-11-16
> **Response to Reviewer #3**
>
> Q1: The main problem is the lack of Novelty and technical contributions. Using a DL-based model to predict the performance is not novel. Many NAS methods using the same trick to estimate the performance of one architecture ahead of running it directly on the hardware.
>
> A1: Thank you for your insightful comments. Yes, the execution time estimation is an important part of NAS. We have done some experiments to show that our work is capable of handling more sophisticated network architectures that are useful of NAS for real applications that require complex models. The existing NAS works have some presumptions/limitations and would not be appropriate for estimating the execution time of a sophisticated model, which is developed with the specific software framework for the hardware platform, e.g., TensorRT for the NVIDIA GPU. For example, the predictor presented in [1] is responsible for identifying a good sub-network under a given trained network and is hard to give an accurate estimate of the execution time of the sophisticated model running on CPU and GPU collaboratively. In addition, in [2], FLOPs is used as one of the metrics for evaluating the neural network model. Nevertheless, it is known that FLOPS is not suitable for estimating the model inference time when the involved operations have non-linear relationships between the number of operations and the incurred execution time. Therefore, regarding the model inference time, it is desired to accurately estimate the model time through a systematic approach, i.e., ResPerfNet.
>
> Q2: Using a DL-base regression is better than normal regression when the sample size is large. The experiment results are not surprising.
>
> A2: We totally agree with your comment on the DL-based and normal regression approaches. However, based on our experiments in this work, we find that the normal regression method, i.e., XGboost, delivers relatively poor performance (MAPE of 29%) when using 80,000 samples to train the XGboost. On the other hand, ResPerfNet requires only 5,000 samples to yield the better MAPE result of 18% for estimating the execution time for a convolutional layer. At the same time, when using 5,000 samples to train XGboost, it produces a high MAPE value of 53%. Hence, in our configuration, it appears that the DL-based approach has the advantage to learn the relationships between the features and the execution time, compared with XGboost.
>
> Q3: Box-cox transformation is a common technique in regression. That is not new.
>
> A3: Thank you for your comment. Our intention is to share our experiences to build the DL-based model predictor among many other design alternatives to construct the predictor, and our experiments show that box-cox transformation outperforms the other transformation schemes when the box-cox transformation is adopted in ResPerfNet.
>
> Q4: Need to prepare the dataset using a lot of samples, i.e. 100000, which is computationally heavy. Require huge storage space to keep the samples for even one platform. Thus the whole method is not efficient. It is hard to generalize it to other hardware/platform
>
> A4: Thank you for your valuable comments. Based on our experiments, 5,000 samples are sufficient to provide a good estimate (MAPE of 18%), which requires 200MB of storage space. 100,000 samples (only 4GB) can further improve the prediction performance (MAPE of 11.78%).
> On the other hand, we use 5TB to keep the TensorRT model variants, apart from the sample data, in order to save the time required for model conversion and optimization to generate the variants needed to train the predictor for other accelerators. In the future, we plan to leverage transfer learning techniques to eliminate the huge space requirement for keeping the variants while our model is able to predict the execution time for different hardware devices.
>
> Q5: Besides, the method only considers some common operations such as conv, pooling, and FC-layer. However, more operations should be considered for example ROI Pooling, NMS, Spatial-to-depth. Those operations are also commonly used in tasks such as detection and SR.
>
> A5: Thanks for the valuable comments. While we are aware that there are still some common operations, our current work focuses on the essential operations forming the well-known models, such as LeNet, AlexNet, and VGG16 so as to validate the effectiveness of ResPerfNet. We believe that with the corresponding sample data, ResPerfNet is able to characterize the above mentioned operations.
>
>
> [1] Han Cai, Chuang Gan, Tianzhe Wang, Zhekai Zhang, and Song Han. Once for all: Train one network and specialize it for efficient deployment. In International Conference on Learning Representations, 2020.
>
> [2] Xiaoliang Dai, Alvin Wan, Peizhao Zhang, Bichen Wu, Zijian He, Zhen Wei, Kan Chen, Yuandong Tian, Matthew Yu, Peter Vajda, Joseph E. Gonzalez. “FBNetV3: Joint Architecture-Recipe Search using Neural Acquisition Function” in arXiv:2006.02049, 2020.

---

> > ### Comment · AnonReviewer3 · 2020-11-18
> > **Thank you for your reply.**
> >
> > Thank you for your reply.
> > I still think using a DL-method to estimate the inference time is a bit trivial. Can you list your technical novelty?

---

> > > ### Author Response · Authors · 2020-11-24
> > > **Response to Reviewer #3**
> > >
> > > A: Thank you for giving us an opportunity to emphasize the novelty of our work.
> > > We aim to tackle the real-world problem: which hardware platform best suits the given DL model, i.e., delivering the best performance, in terms of the model inference time. Nevertheless, we find that it is difficult to achieve the goal with the prior work efficiently.
> > >
> > > As for [1], the averaged scaling parameters for computation and communication are required to give an estimate of the FLOPs delivered by the hardware platform for the target DL model. However, it is hard to anticipate the per-model based averaged scaling parameters on the hardware platform, which highly depends on the software/hardware combination and is hard to obtain such data without physically executing the target DL model on the hardware platform.
> > >
> > > For [2], their proposed method overlooks the communication performance within a given DL model. This limits the applicability of their proposed method, which makes it hard to estimate the inference time when the DL model involves intensive CPU and GPU interactions.
> > >
> > > For [3], their work only predicts fixed layer numbers (at most 25 layers) of DL model performance, fixed batch size, and can not predict the spreading out network case, e,g. inceptionV3, UNet, which do not satisfy the real scenario.
> > >
> > > To the best of our knowledge, the prior work does not provide a viable solution that helps systematically estimate the inference time of a sophisticated DL model on the modern computing hardware, such as GTX 1080 Ti and P1000 GPUs.
> > > Compared with the above works, together with the three-phase performance modeling approach and the proposed residual regression network, ResPerfNet is able to provide accurate estimates of the representative DL models, such as AlexNet and VGG16. Our latest results show that ResPerfNet delivers similar performance on the InceptionV3 model, which is known to present good results on the ImageNet dataset. Last but not least, we believe that ResPerfNet is practical work and we will open source our efforts to the public, in order to help facilitate the development of DL computing systems.
> > >
> > > [1] Hang Qi, Evan R. Sparks, and Ameet Talwalkar. Paleo: A Performance Model for Deep Neural Networks. In International Conference on Learning Representations, 2017.
> > > [2] Daniel Justus, John Brennan, Stephen Bonner, and Andrew Stephen McGough. Predicting the Com-putational Cost of Deep Learning Models.  In International Conference on Big Data, 2018.
> > > [3] Han Cai, Chuang Gan, Tianzhe Wang, Zhekai Zhang, and Song Han. Once for all: Train one network and specialize it for efficient deployment. In International Conference on Learning Representations, 2020.

---

### Official Review · AnonReviewer1 · 2020-10-23
**Reviews**

**Rating:** 4
**Confidence:** 4

**Review:**

Summary: The authors design a specific ResNet for predicting the model execution time on different platforms.

Pros
- conduct extensive experiments, particularly collect a large scale dataset for measuring different architectures, which can be helpful for further works if it can be released publicly

Cons
- The idea is not novel. The main idea is to utilize a ResNet to perform regression on network latency data, which can only be considered as a normal application of ResNet.
- The motivation is questionable. In my opinion, making model execution time prediction more accurate should not be the ultimate end. The proposed ResPerfNet should be applied in network evaluation and search stages in Neural Architecture Search (NAS) area, and validate that a more accurate model performance predictor is helpful for architecture search. But I didn't see any supporting experimental results in this paper.

---

> ### Author Response · Authors · 2020-11-16
> **Response to Reviewer #1**
>
> Thank you for your insightful comments. Yes, the execution time estimation is an important part of NAS. We have done some experiments to show that our work is capable of handling more sophisticated network architectures that are useful of NAS for real applications that require complex models. The existing NAS works have some presumptions/limitations and would not be appropriate for estimating the execution time of a sophisticated model, which is developed with the specific software framework for the hardware platform, e.g., TensorRT for the NVIDIA GPU. For example, the predictor presented in [1] is responsible for identifying a good sub-network under a given trained network and is hard to give an accurate estimate of the execution time of the sophisticated model running on CPU and GPU collaboratively. In addition, in [2], FLOPs is used as one of the metrics for evaluating the neural network model. Nevertheless, it is known that FLOPS is not suitable for estimating the model inference time when the involved operations have non-linear relationships between the number of operations and the incurred execution time. Therefore, regarding the model inference time, it is desired to accurately estimate the model time through a systematic approach, i.e., ResPerfNet.
>
> [1] Han Cai, Chuang Gan, Tianzhe Wang, Zhekai Zhang, and Song Han. Once for all: Train one network and specialize it for efficient deployment. In International Conference on Learning Representations, 2020.
>
> [2] Xiaoliang Dai, Alvin Wan, Peizhao Zhang, Bichen Wu, Zijian He, Zhen Wei, Kan Chen, Yuandong Tian, Matthew Yu, Peter Vajda, Joseph E. Gonzalez. “FBNetV3: Joint Architecture-Recipe Search using Neural Acquisition Function” in arXiv:2006.02049, 2020.

---

### Official Review · AnonReviewer2 · 2020-10-27
**The paper presents a method, based on a residual CNN, for performance prediction of deep neural networks on different hardware platforms. The results are promising, but there is a limited comparison with previous work.**

**Rating:** 5
**Confidence:** 3

**Review:**

The paper presents a method, called ResPerfNet, to predict the performance of deep neural networks. The method relies on a residual neural network that is trained on a large number of different network architectures and performance measures on real hardware.

The paper evaluates the proposed method on three networks, LeNet, AlexNet, and VGG16, in two different frameworks, i.e., TensorFlow and TensorRT. The results are promising, but comparison with other approaches is weak. For example, the proposed method, ResPerfNet, is only compared to one other approach, PerfNet (from a paper that don't seem to be published yet, at least I couldn't find it), using TensorFlow (but not using TensorRT).

The paper has potential to have impact, but it needs to be improved before publication. For example, the following issues need to be addressed:
* Motivation for the selected structure / architecture of ResPerfNet.
* Some confusion about kernels, filters, etc. in the description of the ResPerfNet architecture (Section. 3 + Fig. 1)
* I'm a bit surprised that the dropout layer is very close to the end of the network. Why? Why 0.2 dropout (and not 0.1 or 0.4)?
* It's a bit confusing (and inconsistent) that the index I is left out sometimes and sometimes not, e.g., Eq (2) vs. Eq (3) vs. how it is written in the text flow.
* C(f,d) in Eq (5) is never defined.
* Platform for sample selection / data collection should be mentioned in Section 5.2
* Section 5.4. Although using defining the loss function as MAPLE does reduce the problem with a skewed distribution, it doe not solve it so "cope with it" is a bit strong formulation.
* It is disturbing that one of the main references [Wang 2020] can't be found, despite extensive searching. This is problematic since the only other solution ResPerfNet is compared to is PerfNet, which is published in [Wang 2020]. This limits the possibility to compare this work with previous work. The conference where the [Wang 2020] paper was published took place in mid October 2020, which is after the deadline for ICLR.

---

> ### Author Response · Authors · 2020-11-16
> **Response to Reviewer #2**
>
> Thank you for the valuable comments. We have fixed the equations and contents in the paper according to the comments. The paper will be updated later.
>
> Q: I'm a bit surprised that the dropout layer is very close to the end of the network. Why? Why 0.2 dropout (and not 0.1 or 0.4)?
>
> A: While a dropout layer could be added after a hidden layer, in our study, the dropout layer is mainly used for better and stable prediction results. That is, in some cases, the logarithmic operations in MAPLE will produce erroneous results when unexpected inputs are given, and adding the dropout layer near the end of the network can prevent such a case from happening.
> The dropout ratio of 0.2 delivers the best accuracy (i.e., dropout ratio of 0.2 for MAPE of 11.78%) in our experiments, whereas the ratio of 0.1 leads to MAPE of 11.98% and 0.4 for 12.33%. The ratio can be adjusted accordingly based on the given inputs.
>
> Q: It is disturbing that one of the main references [Wang 2020] can't be found, despite extensive searching. This is problematic since the only other solution ResPerfNet is compared to is PerfNet, which is published in [Wang 2020]. This limits the possibility to compare this work with previous work. The conference where the [Wang 2020] paper was published took place in mid October 2020, which is after the deadline for ICLR.
>
> A: We are sorry for causing the confusion. As [Wang 2020] is published in ACM RACS 2020, which takes place virtually this year, we obtain the results of [Wang 2020] as soon as the preliminary program of ACM RACS 2020 is released online by asking for their paper and research data offline. We find that the work done by [Wang 2020] is very close to this work, and we finally decide to compare with their work in order to present more comprehensive results on the performance estimations of a neural network model and to show the effectiveness of ResPerfNet.

---

> > ### Comment · AnonReviewer2 · 2020-11-16
> > **Thanks**
> >
> > Thanks for the clarifications. It's good that you relate your work to the RACS paper. However, since it wasn't published yet I didn't want to ask for it because that might have compromised the reviewer anonymity.

---

### Decision · Program_Chairs · 2021-01-07
**Final Decision**

**Decision:**

Reject

**Comment:**

This paper presents a new method to predict the performance of deep neural networks. It evaluates the method on three different networks: LeNet, AlexNet, and VGG16 under two different frameworks, TensorFlow and TensorRT.

Reviewer 2 thought that the results were promising but comparison with other approaches was weak (PerfNet being the only baseline). They also asked for motivation for the selected architecture as well as raised a number of points for clarification. R2 was also concerned that the single baseline appeared to have not yet been published. The authors clarified this in their response (it was published in ACM RACS, obtaining results directly from those authors).

Reviewer 1 said that the experiments were extensive, but did not find the approach novel (“a normal application of ResNet”). They suggested NAS as a motivating application rather than stopping at predicting execution time. The authors agreed with the importance of predicting execution time in NAS.

Reviewer 3 agreed with Reviewer 1’s assessment of lacking novelty and technical contribution. They also pointed towards NAS, where many methods are already using neural networks to predict execution time. They were also disappointed by the reduced set of architectural elements considered. The authors responded to R3’s comments, but R3 was still not convinced of novelty.

This looks like a fairly straightforward rejection on the basis of not enough technical merit. The authors are encouraged to explore their approach in the context of NAS as per R1’s suggestion.